

# piRNA-disease association prediction based on multi-channel graph variational autoencoder

Wei Sun[1], Chang Guo[2], Jing Wan[3] and Han Ren[4,5]

[1] School of Information Science and Technology, Qiongtai Normal University, Haikou, China
[2] School of Modern Information Industry, Guangzhou College of Commerce, Guangzhou, China
[3] Center for Lexicographical Studies, Guangdong University of Foreign Studies, Guangzhou, China
[4] Laboratory of Language Engineering and Computing, Guangdong University of Foreign Studies, Guangzhou, China
[5] Laboratory of Language and Artificial Intelligence, Guangdong University of Foreign Studies, Guangzhou, China

## ABSTRACT

Piwi-interacting RNA (piRNA) is a type of non-coding small RNA that is highly expressed in mammalian testis. PiRNA has been implicated in various human diseases, but the experimental validation of piRNA-disease associations is costly and time-consuming. In this article, a novel computational method for predicting piRNA-disease associations using a multi-channel graph variational autoencoder (MC-GVAE) is proposed. This method integrates four types of similarity networks for piRNAs and diseases, which are derived from piRNA sequences, disease semantics, piRNA Gaussian Interaction Profile (GIP) kernel, and disease GIP kernel, respectively. These networks are modeled by a graph VAE framework, which can learn low-dimensional and informative feature representations for piRNAs and diseases. Then, a multi-channel method is used to fuse the feature representations from different networks. Finally, a three-layer neural network classifier is applied to predict the potential associations between piRNAs and diseases. The method was evaluated on a benchmark dataset containing 5,002 experimentally validated associations with 4,350 piRNAs and 21 diseases, constructed from the piRDisease v1.0 database. It achieved state-of-the-art performance, with an average AUC value of 0.9310 and an AUPR value of 0.9247 under five-fold cross-validation. This demonstrates the method's effectiveness and superiority in piRNA-disease association prediction.

# INTRODUCTION

The PIWI-interacting RNAs (piRNAs) are non-coding small RNAs with 24–32 nucleotides (nt) (*Ghosh et al., 2022*), which interact with P-element-induced wimpy testes (PIWI) genes among proteins of the Argonaute family (*Yang, Cho & Zheng, 2020*). Recent research evidence demonstrates that piRNA plays an important role in many biological processes such as maintaining germline DNA integrity (*Wang et al., 2023*), epigenetic regulation (*Sun, Lee & Li, 2022*), transcriptional silencing, and heterochromatin formation

Corresponding author
Han Ren, hanren@gdufs.edu.cn

(*Zhao et al., 2019*). Biological experiments also show that piRNA becomes a significant biomarker for the diagnosis of diseases thanks to the abnormal expression of piRNAs and PIWI proteins in various diseases (*Weng, Li & Goel, 2019*). For example, piR-38240, piR-25783, piR-40666, and piR-28467 are valuable to the diagnosis of Alzheimer's disease (*Olufunmilayo & Holsinger, 2023*); piR-54265 level is positively correlated with tumor tissue level in the serum of human with colorectal cancer from the research of *Mai et al. (2020)*; the level of piR-26399 in the serum of male subfertility patients was significantly higher than in healthy males as discovered by *Kumar et al. (2019)*; piR-26399 could be used as a potential biomarker of male subfertility; piR-823 plays a vital role in many cancer diseases (*Maleki Dana, Mansournia & Mirhashemi, 2020*; *Wang et al., 2021*, *2022b*; *Wei, Ding & Liu, 2020*). Therefore, finding disease-related piRNAs will contribute to the diagnosis of these diseases.

Currently, a number of piRNAs have been discovered, and large-scale data sources related to piRNA, such as piRNAdb (*Piuco & Galante, 2021*), piRNABank (*Sai Lakshmi & Agrawal, 2008*) and piRBase (*Wang et al., 2022a*, *2019a*), have been established, Such datasets provide abundant information related to piRNA, including genomic element, physicochemical property, and the sequence information. Databases of piRNA-disease association verified by biological experiments have also been built. For example, *Muhammad et al. (2019)* built the database of piRDisease by collecting 7,939 experimentally verified piRNA-disease associations with 4,796 piRNAs and 28 diseases from more than 2,500 published biomedical literature; *Zhang et al. (2020)* built a database named piRPheno through a manual extract method on opened publications to obtain the association between piRNAs and diseases. It contains 9,057 experimentally verified associations between 474 piRNAs and 204 diseases. On the other hand, building such data sources is always costly and time-consuming with complex biological experiments (*Chen et al., 2019*; *Ernst, Odom & Kutter, 2017*). To address the problem, computational methods for identifying piRNA-disease association have been put forward continuously in recent years. One type of research is to treat the association prediction as a classification task. For instance, *Zheng et al. (2020a)* introduced a computational model named APDA based on Random Forest (RF) to find disease-related piRNAs. In their model, two groups of comparative experiments were set to study the impact of features on prediction performance. One group obtained the feature representation of piRNA and disease through collaborative filtering (CF), while another group measured the similarity of piRNA sequence as well as the disease semantics for feature construction. The experiment on the benchmark dataset piRDisease V1.0 The experiment on the benchmark dataset piRDisease V1.0 (*Muhammad et al., 2019*) showed that the latter one outperformed the former one, indicating that those features was effective to the prediction task. *Wei, Xu & Liu (2021)* investigated the impact of the high-quality negative samples to the prediction task. They also proposed a computational method called iPiDi-PUL, which used positive unlabeled learning (PUL) (*Claesen et al., 2015*; *Mordelet & Vert, 2014*) to identify the potential association between piRNA and disease. Such problem can also be viewed as a link prediction or recommendation task, where piRNAs related to a disease are always ranked for choosing. For example, *Zhang, Hou & Liu (2022)* developed iPiDA-LTR, a

learning To rank (LTR) model, to identify the potential association between piRNA and disease. It uilitized gradient increasing decision tree to acquire feature representations of piRNAs and diseases, and then employed multiple models, *i.e.*, Random Forest and support vector machine (SVM), to compute the association score between the unknown piRNA and each known disease.

In recent years, more and more deep learning computational methods have been introduced into this research field. *Wei, Ding & Liu (2020)* utilized a convolutional neural network (CNN) model to get features of piRNA and disease and adopt a two-step PUL method for association prediction. At the first step, an SVM is employed to get high-quality negative samples, while these samples and known positive samples were fed into another SVM classifier to predict potentially disease-related piRNAs in the second step. *Ali, Tayara & Chong (2022)* utilized one-hot encoding method to encode piRNA sequences and disease semantics, and employed CNN to learn feature representation. In addition, some researchers also use graph embedding methods to obtain the features of the piRNA-disease association network. For example, *Zheng et al. (2020b)* proposed to utilize graph attention network (GAT) (*Veličković et al., 2017*) to learn the feature representation of piRNAs and diseases. *Hou, Wei & Liu (2022)* introduced a multiple-view learning framework by graph convolution network (GCN) (*Kipf & Welling, 2016a*). They constructed two GCN models, one was Asso-GCN, which learned feature representation of association information from heterogeneous nodes in the association network, and the other was Sim-GCN, which captured similarity features from homogeneous nodes of piRNA or disease. The experimental results showed that graph models helped capture non-linear association information over piRNA-disease association networks.

On the other hand, although existing graph-based methods achieve good performance, they still have limitations. Firstly, it is challenging to fuse multi-layer similarity information between piRNAs and between diseases. Current methods always combined similarity features with an average operation (*Wei, Xu & Liu, 2021*; *Ji et al., 2021*), which may be detrimental to those important association information of piRNAs and diseases. Secondly, current representation learning methods always employ sophisticated neural networks such as CNN and GAT, which requires rich labeled data, whereas piRNA data are sparse and imbalanced. In other words, such models are difficult to get underlying feature representation by leveraging on unknown association information in graphs.

In this article, a novel model named the Multi-Channel Graph Variational Autoencoder (MC-GVAE) is proposed to predict potential piRNA-disease associations. The motivation lies in two folds: (1) the graph VAE model learn feature representation in an unsupervised manner (*Kipf & Welling, 2016b*), which helps get underlying relationships in sparse piRNA-disease association networks; (2) the multi-channel method helps fuse multiple similarity information. Specifically, MC-GVAE first constructs four similarity networks by measuring sequence or semantic similarity and GIP kernel similarity of piRNA and disease (*Van Laarhoven, Nabuurs & Marchiori, 2011*). Then, the graphs are modeled *via* a graph VAE to obtain deep feature representations of piRNAs and diseases, and the homogeneous similarity networks are integrated by a multi-channel method, where parameters are shared over different graph VAEs for homogeneous similarity networks. Finally, the

feature representations are concatenated and fed into a full-connected neural network to prediction. The contributions of this article are listed as follows:

1) This article adopts a graph variational autoencoder (graph VAE) model to learn the feature representation of piRNA and disease, trying to capture the data distribution features in the graph.

2) This article proposes a multi-channel method to fuse information in homogeneous similarity networks in order to better acquire the relations in piRNAs and diseases, respectively.

3) Computational experiments show that the state-of-the-art performance for piRNA-disease association prediction is achieved by this model.

## MATERIALS AND METHODS

### Benchmark datasets

The benchmark dataset is constructed from piRDisease v1.0 (*Muhammad et al., 2019*), which is the same version used by iPiDi-PUL (*Wei, Xu & Liu, 2021*), iPiDA-sHN (*Wei, Ding & Liu, 2020*), and DFL-PiDA (*Ji et al., 2021*), which offers the information of piRNA-disease associations. There are 7,939 associations in this database, labeled by manual work. In this study, we used the same data preprocessing steps as iPiDA-sHN (*Wei, Ding & Liu, 2020*). These steps include filtering redundant and non-human piRNAs, removing piRNAs that have no sequence information in the database. After applying these steps, we obtained a benchmark dataset that is identical to the datasets used by iPiDi-PUL (*Wei, Xu & Liu, 2021*), iPiDA-sHN (*Wei, Ding & Liu, 2020*), and DFL-PiDA (*Ji et al., 2021*), containing 5,002 experimentally validated associations with 4,350 piRNAs and 21 diseases. The benchmark dataset is a publicly accessible resource at http://bliulab.net/iPiDi-PUL/dataset/.

### Computation method

This study proposes a MC-GVAE model to predict the potential associations between piRNAs and diseases. The model is a pipeline with four sequential steps: data preprocessing, similarity network construction, representation learning based on Multi-channel graph VAE, and prediction. In the step of data preprocessing, the piRNA sequence expression and the adjacency matrix of piRNA-disease associations are extracted from the piRDisease V1.0 database. The adjacency matrix is as follows:

$$A = \begin{bmatrix} a_{1,1} & a_{1,2} & \ldots & a_{1,n} \\ a_{2,1} & a_{2,2} & \ldots & a_{2,n} \\ \vdots & \vdots & \ddots & \vdots \\ a_{m,1} & a_{m,2} & \ldots & a_{m,n} \end{bmatrix} \tag{1}$$

where $a_{i,j} = 1$ if the *i*-th RNA is associated with the *j*-th disease, otherwise $a_{i,j} = 0$. Then, the k-mer algorithm is used to calculate the sequence features of each piRNA, and the Pearson product-moment correlation coefficient is used to obtain the sequence similarity

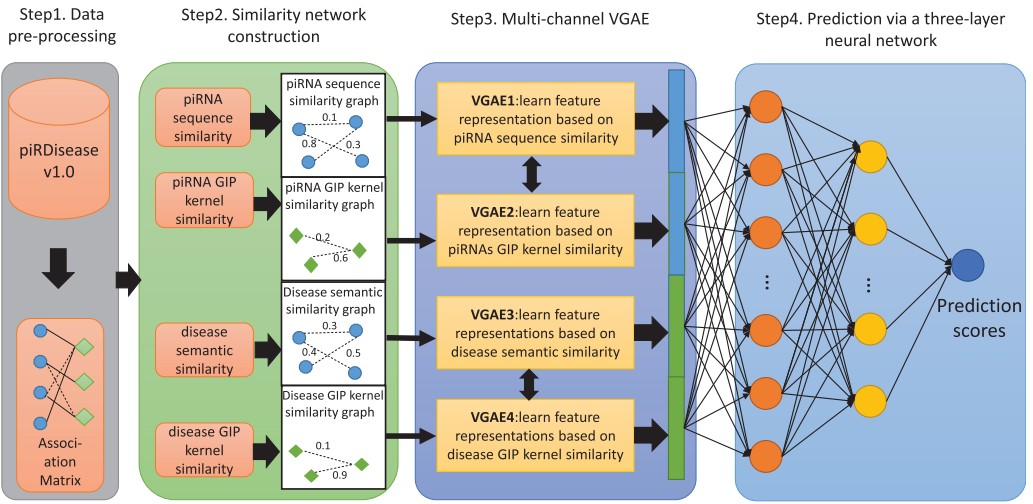

**Figure 1** **The flowchart of MC-GVAE.** The diagram illustrates the core processes of the proposed MC-GVAE model. In the diagram, blue dots represent piRNA entities, and green diamonds represent disease entities. In Step 1, data is pre-processed from the piRDiseaseV1.0 database to form an association Matrix. Step 2 involves constructing four similarity networks based on piRNA sequence similarity, piRNA GIP kernel similarity, disease semantic similarity, and disease GIP kernel similarity. Step 3 introduces the Multi-Channel Graph Variational Autoencoder, where graph VAE learns feature representations in an unsupervised manner and integrates various similarity networks using a multi-channel approach. In Step 4, the feature representations are put into a three-layer neural network for predicting piRNA-disease associations.                               

matrix of piRNAs. Meanwhile, based on the adjacency matrix of piRNA-disease associations, the Gaussian kernel function is used to calculate the GIP similarity matrix of piRNAs and diseases. In addition, the directed acyclic graph (DAG) of each disease in the associated matrix is obtained from the disease descriptor vocabulary MeSH, and the semantic value of each disease is calculated, resulting in the semantic similarity matrix of diseases. In the step of similarity network construction, four different similarity subnetworks are constructed based on the four different similarity features obtained in the previous step. In the step of representation learning, four identical VGAE models are used to separately learn the feature representations of piRNAs and diseases from four different similarity subnetworks, and then concatenate the four different feature representations to better mine the associations between piRNAs and diseases. In the step of prediction, a three-layer fully connected neural network is used to classify and predict all sample pairs, obtaining the final prediction results.The flowchart of MC-GVAE's computation process is shown in Fig. 1. The following sections will give a detailed introduction to each step's specific implementation.

## Similarity network construction

To construct the similarity network, we computed the similarity features between piRNAs and diseases from four different perspectives, namely piRNA sequence similarity, disease semantic similarity, piRNA GIP kernel similarity, and disease GIP kernel similarity. These similarity features can reflect the different correlation patterns between piRNAs and

diseases, which can help our model to better mine the associations between piRNAs and diseases.

### piRNA sequence similarity network

piRNA sequence similarity network is an undirected graph, where each node represents a piRNA sequence and each edge represents the relation of two nodes. Generally, the relation can be measured by the similarity of features of two sequences. One sophisticated method to obtain the features of piRNA sequences is K-mer (*Li et al., 2016*), an algorithm to extract sequence-derived features using the overlapping shifting window. In this study, K-mer is utilized to divide RNA nucleotide sequences into subsequences of length ð and count their occurrence frequencies (*Kirk et al., 2018*), to obtain the sequence feature vectors of RNA. Following the algorithm of *Ji et al. (2021)*, the value of K was set to 3 in K-mer to obtain a 64-dimensional vector.

Pearson product-moment correlation coefficient is employed in this study to calculate the similarity of piRNA sequence features. The calculation method is as follows:

$$P\left(p_i, p_j\right) = \frac{Cov\left(F_{P_i}, F_{P_j}\right)}{\sigma_{F_{P_i}} \sigma_{F_{P_j}}} \tag{2}$$

where $F_{P_i}$ and $F_{P_j}$ denotes the sequence feature vector of i-th and j-th piRNA, respectively. $Cov\left(F_{P_i}, F_{P_j}\right)$ denotes covariance between $F_{P_i}$ and $F_{P_j}$. $\sigma_{F_{P_i}}$ and $\sigma_{F_{P_j}}$ denotes the standard deviation between $F_{P_i}$ and $F_{P_j}$.

### Disease semantic similarity network

In this study, the disease semantic similarity is computed *via* the disease database MeSH (*Lipscomb, 2000*). The knowledge of disease in such database can be organized as a directed acyclic graph $G = (V, E)$, where V is the set of nodes represent diseases and E the set of edges represent the relations of diseases. The semantic contribution value $D_d(t)$ of the disease t to the disease d is measured as follows:

$$D_d(t) = \begin{cases} 1, & \text{if } t = d \\ max\{\alpha * D(t'), t' \in children\ of\ t\}, & \text{if } t \neq d \end{cases} \tag{3}$$

where $\alpha$ denotes the semantic contribution factor, with a value of 0.5 (*Wang et al., 2010*). t denotes a disease node in the node set V of the directed acyclic graph, and t' denotes a child node of t. For example, in the directed acyclic graph of Brain Neoplasms, Central Nervous System Disease is a subclass of Brain Diseases. Therefore, Central Nervous System Disease represents t' and Brain Diseases represents t in this case.

The disease semantic similarity $D(d_i, d_j)$ is based on an assumption that the more DAGs shared by disease pairs, the larger their similarity (*Wang et al., 2010*). The similarity is defined as follows:

$$D\left(d_i, d_j\right) = \frac{\sum_{t \in T(d_i) \cap T(d_j)}\left(D_{d_i}(t) + D_{d_j}(t)\right)}{D(d_i) + D\left(d_j\right)} \tag{4}$$

where $D(d_i)$ and $D(d_j)$ denotes the semantic value of disease $d_i$ and $d_j$, respectively, and can be obtained as follows:

$$D(d) = \sum_{t \in T(d)} D_d(t) \tag{5}$$

### GIP kernel similarity networks

Experimental evidence shows that similar piRNAs usually show similar correlation patterns with diseases (*Chen et al., 2016*). On the other hand, such patterns always mean nonlinear relationship between piRNAs and diseases. Therefore, The GIP kernel similarity was utilized to measure the association of piRNAs and diseases, based on the topology of the known association knowledge of piRNAs and diseases and the interaction profile corresponding to each piRNA and disease (*Van Laarhoven, Nabuurs & Marchiori, 2011*). In other words, piRNAs having higher GIP kernel similarities with those disease-related ones could be the positive candidates in piRNA-disease association prediction. The GIP kernel similarity is defined as follows:

$$P = \exp\left(-\lambda_p \parallel A(p_i) - A(p_j) \parallel^2\right) \tag{6}$$

$$D = \exp\left(-\lambda_d \parallel A(d_i) - A(d_j) \parallel^2\right) \tag{7}$$

$$\lambda_p = \frac{1}{N_P} \sum_{m=1}^{N_P} \parallel T(p_m) \parallel^2 \tag{8}$$

$$\lambda_d = \frac{1}{N_d} \sum_{n=1}^{N_d} \parallel T(d_n) \parallel^2 \tag{9}$$

where $A(p_i)$ denotes the association between piRNA $p_i$ and all diseases in the sample. $A(d_i)$ denotes the association between disease $d_i$ and all piRNA in the sample. $N_p$ and $N_d$ denotes the number of piRNA and disease in the sample, respectively.

## Representation learning based on multi-channel graph VAE

A graph VAE is a kind of variational autoencoder applied to graphs composed of GCN and VAE and is an unsupervised learning framework. Therefore, the graph VAE not only learns graph structure features but also generates data distribution features of piRNA and disease. In the section that follows, GCN and graph VAE will be briefly introduced.

### GCN

Graph convolution network (GCN) is a powerful representation learning method in graph structure data. Here a GCN model was employed to extract four feature vectors of piRNA and disease from four similarity networks, including piRNA sequence, disease semantics, piRNA GIP kernel, and disease GIP kernel similarity networks. Firstly, an adjacency matrix was constructed with the weight based on piRNA sequence similarity. Then, the piRNA-disease association matrix $A \in \mathrm{R}^{4,350 \times 21}$ is set as graph features, in which each row of the piRNA-disease association matrix is initialized as each piRNA feature $H$. Finally, a

two layers GCN is utilized to obtain a low-dimensional feature matrix. $H^i$ and $W^i$ denotes the node feature and the parameter matrix of the $i$-th layer. Therefore, the node features vectors $H^{l+1}$ in each layer is represented as follows:

$$H^{l+1} = ReLu\left( \tilde{D}^{-\frac{1}{2}} \tilde{A} \tilde{D}^{-\frac{1}{2}} H^l W^l \right) \tag{10}$$

where the ReLu() function serves as an activation mechanism to introduce non-linearity into the model.

### Graph VAE

The graph VAE model is an end-to-end framework, including an encoder and a decoder. In the encoder stage, a GCN is employed to learn the feature representation vector. In the decoder one, the similarity matrix was reconstructed by calculating the inner product of the vector.

The encoder contains two layers. The first layer is to obtain the low dimensional vector of each node, while the second layer generates its distribution feature. The two layers are formalized as follows:

$$\mu = GCN_\mu(X, A) \tag{11}$$
$$\log \sigma = GCN_\sigma(X, A) \tag{12}$$

where $GCN_\mu()$ and $GCN_\sigma()$ are two GCNs and share the weight, and $X$ is the feature matrix of nodes. The latent representation $Z$ is defined as follows:

$$Z = \mu + \sigma * \varepsilon \tag{13}$$

where $\varepsilon \sim N(0, 1)$.

Optimal parameters were obtained by reconstructing the piRNA similarity matrix through the calculation of the inner product between the feature embedding $F$ and its transpose $F^T$. During the learning time, consideration was given to two loss functions, binary cross-entropy and KL-divergence, to minimize the distance between the target and the reconstructed matrix. The loss function $\mathcal{L}$ is denoted as follows:

$$\mathcal{L} = \mathbb{E}_{q(Z|A,X)}[log p(A|Z)] - KL[q(Z|A, X)||p(Z)] \tag{14}$$

### Multi-channel representation learning

Essentially, the piRNA sequence similarity network and the piRNA GIP similarity network demonstrate the relevance of piRNAs with different perspective. That is the same situation between the disease semantic similarity network and the disease GIP similarity network. Therefore, A tensor can be used to represent such similarity networks, each slice denoting one kind of relevance meaning. A multi-channel feature representation model can also be built to learn from such tensor network (*Chen et al., 2022*). The aim of the multi-channel method is to model information from multiple aspects, which may lead to sufficient feature representation learning. Average pooling, parameter sharing and cross-attention are always leveraged for multi-channel modeling. In this study, a multi-channel learning

model is built, where the module of each channel learns the representation of piRNA by leveraging on one similarity network respectively.

## Predicting piRNA-disease association based on neural network

Firstly, the representation of piRNA-disease association pairs is obtained by concatenating four features $Z_{PS}$, $Z_{PG}$, $Z_{DS}$, and $Z_{dG}$, where $Z_{PS}$ denotes the distribution representation learned from the piRNA sequence similarity network, $Z_{DS}$ from the disease semantic similarity network, $Z_{PG}$ from the piRNA GIP kernel similarity network, and $Z_{dG}$ from the disease GIP kernel similarity network:

$$F = (Z_{PS}, \ Z_{DS}, Z_{PG}, \ Z_{dG}) \tag{15}$$

Then, the final representation $F$ was fed into a three layers full connection neural network classifier, and get the prediction score $\hat{y}_i$ as follows:

$$\hat{y}_i = sigmoid(MLP(F)) \tag{16}$$

where $sigmoid()$ is an activation function to transform $\hat{y}_i$ to be a number between 0 and 1. In this study, the piRNA-disease association pairs with $\hat{y}_i > 0.5$ as 1 (positive sample). The otherwise labeled as 0 (negative sample). However, unlike graph VAE, only the binary cross-entropy was utilized as loss function for prediction.

## RESULTS

### Experimental setup and hyperparameter configuration

In the experiments, a five-fold cross-validation is conducted in the training stage in order to make this model more reliable. More specifically, samples in the training data are divided equally into five subsets, four of which are utilized for training and one of which for testing. Note that each subset has one chance to be the test data. The evaluation metrics that are mostly employed in association prediction tasks, including accuracy, recall, precision, F-1 score, and Matthews correlation coefficient (MCC).

To visually illustrate the performance, this study employs the receiver operating characteristic (ROC) curve and the precision-recall (PR) curve. The ROC curve reflects the relationship between the true positive rate and the false positive rate of the model at different thresholds, while the PR curve reflects the relationship between the precision and recall of the model at different thresholds. In addition, the Area Under Curve (AUC) and the area under PR curve (AUPR) were also employed as evaluation metrics, which are frequently used to measure the performance of classification or prediction tasks. AUC and AUPR are obtained by calculating the area under the ROC curve and the PR curve, respectively, and they can comprehensively reflect the performance of the model at different thresholds.

The experiment is based on the Python deep learning framework PyTorh v1.11.0, and the GCN model is performed by adopting Deep Graph Library (DGL) v0.8.1, a highly-performant package for graph neural networks (*Wang et al., 2019b*). The computational method of this article was performed on the Ubuntu 20.04 with two 1080Ti GPUs. Moreover, several important hyperparameters settings of this study were presented.

**Table 1 Model hyperparameter settings.**

| Component | Hidden1 dimension | Hidden2 dimension | Output dimension | Learning rate | Loss function | Activation function |
|---|---|---|---|---|---|---|
| GVAE | 48 | 16 | – | 0.001 | Weighted sum of binary cross-entropy and KL divergence | ReLU |
| Classifier | 48 | 16 | 1 | 0.001 | Weighted sum of binary cross-entropy | Sigmoid |

**Table 2 The performance of MC-GVAE based on five-fold CV.**

| Test fold | AUC | AUPR | Accuracy | Precision | Recall | F1-score | Mcc |
|---|---|---|---|---|---|---|---|
| 1 | 0.9331 | 0.9286 | 0.9005 | 0.9339 | 0.8534 | 0.8945 | 0.8042 |
| 2 | 0.9237 | 0.9180 | 0.9010 | 0.9356 | 0.8632 | 0.8979 | 0.8047 |
| 3 | 0.9303 | 0.9252 | 0.8926 | 0.9401 | 0.8403 | 0.8874 | 0.7898 |
| 4 | 0.9351 | 0.9340 | 0.9130 | 0.9480 | 0.8708 | 0.9077 | 0.8285 |
| 5 | 0.9295 | 0.9177 | 0.9030 | 0.9315 | 0.8727 | 0.9011 | 0.8078 |
| Average | 0.9310 | 0.9247 | 0.9020 | 0.9390 | 0.8601 | 0.8977 | 0.8070 |

According to the previous work (*Kipf & Welling, 2016b*), GCN was also adopted as an encoder, and the experiment demonstrates that if the number of hidden layers is over 2, the prediction performance of the model will be declined. The dimension of the vector in hidden layer1 and hidden layer2 were set as 48 and 16, and the learning rate is determined as 0.001. The training epoch is detected as 50, and the Adam optimizer and loss function with cross entropy and KL divergence were employed to update the parameters of MC-GVAE in each epoch. The following Table 1. provides a detailed overview of the model architecture and hyperparameter settings for GVAE and the classifier.

### Predicting piRNA-disease association based on neural network

A benchmark dataset (*Wei, Xu & Liu, 2021*) based on the piRDisease database v1.0 was made use of. The benchmark dataset included 10,004 piRNA-disease association pairs, in which the positive sample pairs contained 5,002 associations with 4,350 piRNAs and 21 diseases. In addition, 5,002 negative sample pairs were randomly selected from unknown piRNA-disease association sample pairs. Finally, the detail of the results were shown in Table 2, and the results of AUC and AUPR are shown in Figs. 2 and 3, respectively. Table 2 shows the average performance of model MC-GVAE on the benchmark dataset based on five-fold cross-validation.

Table 2 shows that MC-GVAE achieved high levels of performance on all metrics, with average values of 0.9310, 0.9247, 0.9020, 0.9390, 0.8601, 0.8977, and 0.8070, respectively. This indicates that MC-GVAE can effectively learn the complex features of piRNA and disease, and use the graph structure information to improve the prediction accuracy. From the description of Fig. 2, the ROC curve is close to the upper left corner, the P-R curve is close to the upper right corner, and the curves of each fold are very close, which shows that

_Peer_J Computer Science

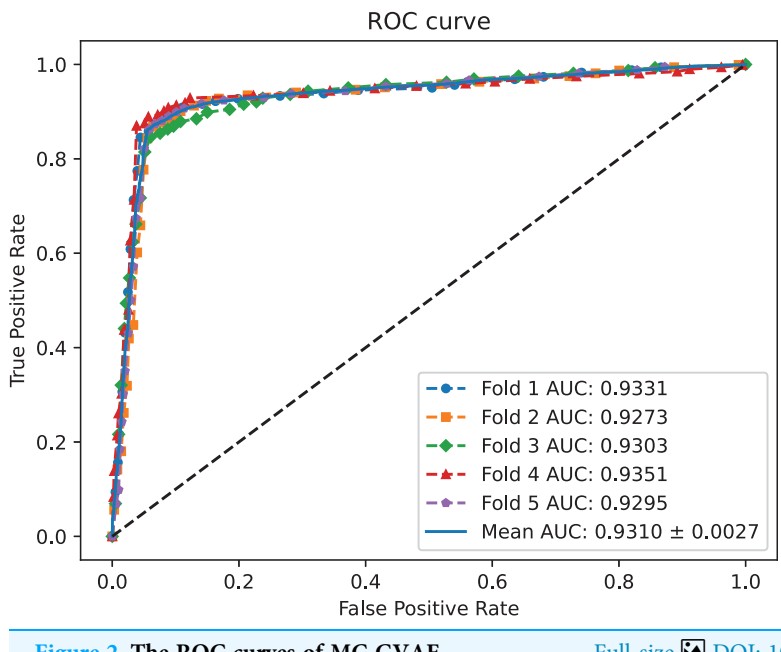

**Figure 2  The ROC curves of MC-GVAE.**               

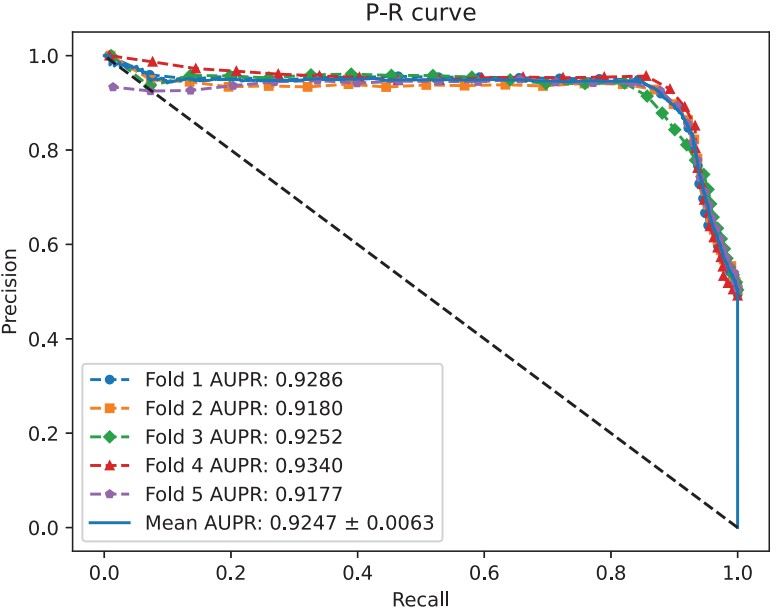

**Figure 3  The P-R curves of MC-GVAE.**               

MC-GVAE has excellent accuracy and robustness in distinguishing positive and negative samples and precisely retrieving positive samples.

## The effect of GCN encoding module

In order to explore the impact of different encoders, we make a comparative experiment, where two encoders are built: one is based on a GCN model, and the other is implemented with a linear layer neural network. The decoder and the training objective are kept

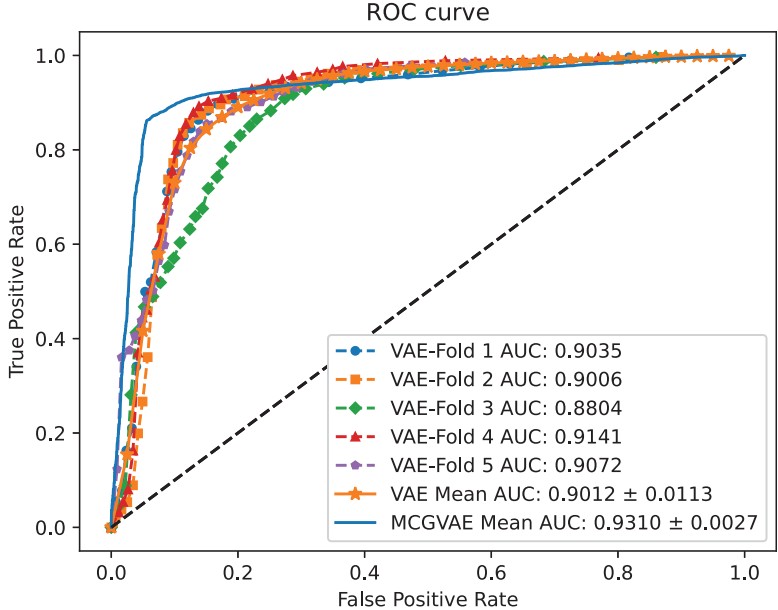

**Figure 4** **The ROC curves of MC-GVAE and VAE.**

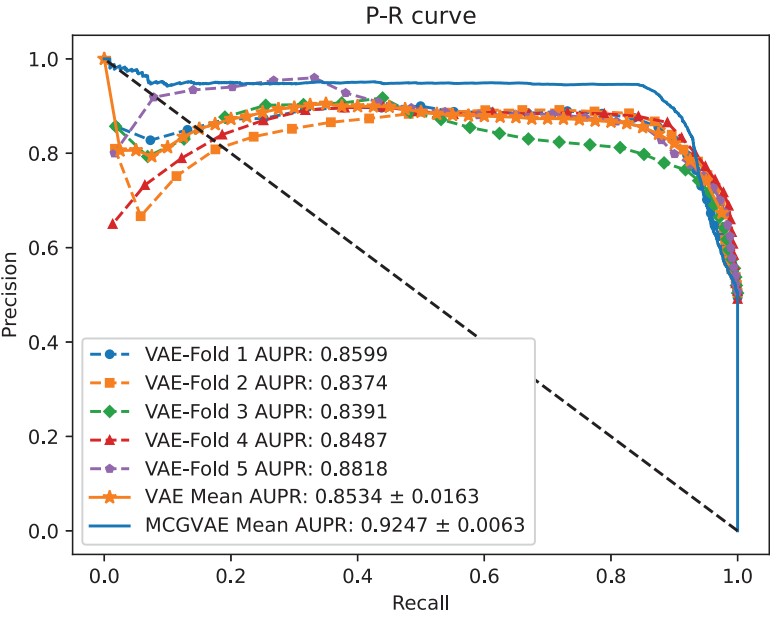

**Figure 5** **The P-R curves of MC-GVAE and VAE.**

unchanged. AUC and AUPR scores of the two models VAE and MC-GVAE on the benchmark dataset are shown in Figs. 4 and 5, respectively.

It can be seen from the Figs. 4 and 5 that the AUC and AUPR scores of MC-GVAE are higher than those of VAE, respectively, under the same benchmark dataset based on 5-fold cross validation, which also means that the prediction performance of MC-GVAE is significantly better than that of MC-GVAE. It indicates that the GCN encoder can

**Table 3 Performance of graph representation learning models.**

| Models | AUC | AUPR | Acc | Pre | Recall | F1 | Mcc |
|---|---|---|---|---|---|---|---|
| GCN | 0.6508 | 0.7659 | 0.5741 | 0.2793 | 0.3757 | 0.6508 | 0.3107 |
| Graphormer | 0.6063 | 0.6469 | 0.6517 | 0.6645 | 0.4989 | 0.5393 | 0.3081 |
| GAT | 0.8971 | 0.9034 | 0.7852 | 0.9309 | 0.6245 | 0.7267 | 0.6146 |
| GVAE | 0.9310 | 0.9247 | 0.9020 | 0.9390 | 0.8601 | 0.8977 | 0.8070 |

effectively obtain the structure information of four views graph, and have an important impact on MC-GVAE model.

## Performance comparison of graph representation learning

To investigate the performance of different graph neural networks in modeling the similarity knowledge with four similarity networks, three models, GCN, GAT and Graph Transformer model—Graphormer (*Ying et al., 2021*), were employed. A two-layer GCN model was built to obtain low-dimensional feature representations of each view, and the numbers of the hidden units in the first and the second layer were 48 and 16, respectively. A single-layer GAT with an eight-head self-attention mechanism was also applied to learn the feature representations of four different views. Furthermore, the Graphormer model was also applied to learn feature representations, utilizing the GraphormerLayer provided in the DGL library. The hidden layer was designed with 48 units, num_heads set to 3, and a Dropout rate of 0.1 for the feature layer.

The number of parameters in all models is the same in the hidden layer, and the dimension of the input and the output vector were set to 48 and 16, respectively. The learning rate of the model is 0.001 during the training process.

Using a five-fold cross-validation method, four GNN models were trained on the benchmark dataset and their prediction performance was compared using common evaluation indicators, including AUC, AUPR, precision, recall, F1-scores, and MCC, for the purpose of comparing the performance of representation learning. Finally, the average of these indicators was listed as experimental results in Table 3.

Experimental results indicate that the graph variational autoencoder (GVAE) model surpasses other models across all evaluation metrics, particularly in terms of area under the curve (AUC), area under the precision-recall curve (AUPR), precision, recall, F1 score, and MCC. These findings affirm the efficacy and superiority of the graph VAE in the task of predicting piRNA-disease associations. Although the Graphormer, as a Graph Transformer model, demonstrates potential abilities in processing graph-structured data, it remains inferior to the GVAE models, which is built for capturing complex relations over graphs. Moreover, the generative learning approach of the graph VAE offers enhanced interpretability in understanding the associations between piRNAs and diseases. Consequently, this research underscores the significance of graph VAE in such tasks and provides new directions for future studies.

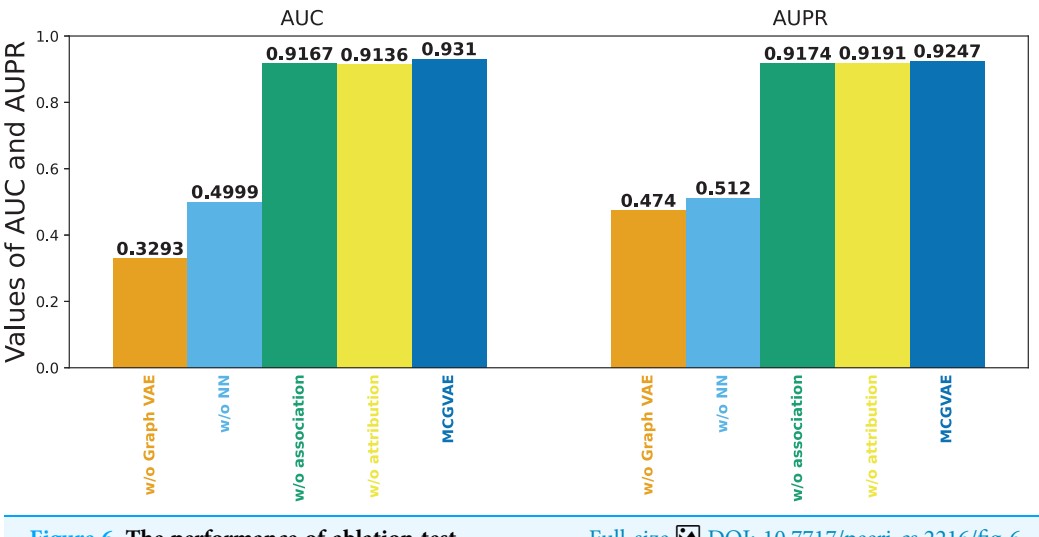

**Figure 6** The performance of ablation test.

## Ablation study

The feature representation of two different views was concatenated as the final classification feature of piRNA-disease association pairs in this model. In order to explore the contribution of the features of two views to the overall performance of MC-GVAE, an ablation study was performed under the benchmark dataset on five-fold cross validation. Some variants of MC-GVAE model were as follows:

- MC-GVAE without graph VAE: the graph VAE is removed, and four similarity features are directly fed into a three-layer full-connected neural network to predict the piRNA-disease association.
- MC-GVAE without NN: it only uses the inner product to reconstruct two prediction matrices based on the features of two views and gets the final prediction matrix by averaging two re-construction matrices.
- MC-GVAE without association: only two similarity calculation data, piRNA sequence and disease semantics, are employed to build the input data to learn node features of piRNA and disease.
- MC-GVAE without attribution: only the GIP similarity data for piRNA and disease are employed as the input data to learn node features of piRNA and disease.

Figure 6 shows the results of the models mentioned above. It can be seen that the original model still achieves the best performance, while MC-GVAE without graph VAE has the lowest AUC and AUPR performance. It can also be seen that, the performance of MC-GVAE without association view and MC-GVAE without attribution view based on different views by using the same computing framework is similar, and both achieve remarkable results. In addition, the best performance based on two different view features was achieved by this model MC-GVAE.

To quantify the impact of each component, statistical significance tests were conducted for the four variants and MC-GVAE. As demonstrated by the T-test results in Table 4, the

**Table 4 Significance testing results for models of ablation test.**

| Variant | Statistic | *p*-value | Interpretation |
|---|---|---|---|
| w/o Graph VAE | 61.905 | 0.000 | Significant decrease, indicating the importance of the Graph VAE component. |
| w/o NN | 68.509 | 0.000 | Significant decrease, indicating the importance of the NN component. |
| w/o Association | 0.849 | 0.396 | No significant decrease, suggesting the association view may not be critical. |
| w/o Attribution | 0.382 | 0.703 | No significant decrease, suggesting the attribution view may not be critical. |

MC-GVAE without graph VAE component and the MC-GVAE without NN component resulted in a significant decrease in performance, with statistical values of 61.905 and 68.509, respectively, and *p*-values of 0.000, indicating the importance of these two components to the model's performance. In contrast, the variants without association and without attribution did not show statistically significant performance decreases, with statistical values of 0.849 and −0.382, respectively, and *p*-values of 0.396 and 0.703.

These results underscore the significance of the graph VAE and n three-layer full-connected neural network components in our model. The original MC-GVAE model achieved optimal performance in integrating feature representations from two different views, demonstrating that our proposed multi-view computational framework can significantly enhance the performance of piRNA-disease association prediction tasks.

## Comparison with other methods

In this part, the performance of MC-GVAE and comparative methods including iPiDi-PUL (*Wei, Xu & Liu, 2021*), iPiDA-sHN (*Wei, Ding & Liu, 2020*), and DFL-PiDA (*Ji et al., 2021*), based on the same benchmark dataset under five-fold CV.

- iPiDi-PUL is a widely used and cited method for piRNA-disease association prediction, which also adopts four types of features like our method. Specifically, it adopts an ensemble learning method, which employs multiple random forests with different depths to predict piRNA-disease association. It extracts key features by principal component analysis (PCA) (*Abdi & Williams, 2010*), and gets various prediction scores *via* multiple random forests classifiers. Finally, averaging all classifier scores is used by the model. Different from our study, it does not use a graph-based learning model to capture the complex relationships between piRNAs and diseases, but relies on PCA to extract features. Choosing iPiDi-PUL as a comparative method may help to study the performance of graph-based learning models against models of reduction dimensions in representation learning.
- iPiDA-sHN is a state-of-the-art method for piRNA-disease association prediction. It is a two-step PUL model. It trains the initial classifier with positive samples and randomly selected negative samples, and then trains another classifier using positive samples and negative samples acquired in the previous step. Support Vector Machine (SVM) is leveraged to build the classifier. Different from our method, it does not use a graph variational autoencoder, but a CNN, to learn the latent representations of piRNAs and diseases. Choosing iPiDA-sHN as a comparative method may help to study the

**Table 5 The performance comaprison of MC-GVAE and other methods.**

| Methods | Average AUC | Average AUPR |
|---|---|---|
| iPiDi-PUL (*Wei, Xu & Liu, 2021*) | 0.8540 | 0.7852 |
| iPiDA-sHN (*Wei, Ding & Liu, 2020*) | 0.8867 | 0.8340 |
| DFL-PiDA (*Ji et al., 2021*) | 0.9042 | – |
| MC-GVAE | 0.9310 | 0.9247 |

performance of graph variational autoencoder against other graph-based learning models.

- DFL-PiDA is constructed from a convolutional denoising autoencoder neural network combined with an extreme learning machine (ELM) to predict disease-related piRNAs. It utilizes a single hidden layer feedforward neural network to build the ELM. Different from our method, it use a convolutional denoising autoencoder to learn feature representations. Choosing DFL-PiDA as a comparative method may help to study the performance of graph variational autoencoder against other autoencoder models.

Our MC-GVAE method was performed on the Ubuntu 20.04 with two 1080Ti GPUs environment. The results are shown in Table 5. We can see that, the MC-GVAE model outperforms other methods in terms of both average AUC and AUPR metrics, demonstrating that it can effectively predicts piRNA-disease associations and has higher prediction accuracy and stability. Meanwhile, there are some possible reasons of low performance of the three models: (1) iPiDi-PUL employs PCA to extract key features, which may overlook some relevant information; (2) iPiDA-sHN adopts SVM to construct the classifier, which may be influenced by the sample imbalance; (3) DFL-PiDA merely fuses four similarity matrices, which fails to learn the rich feature representations of piRNA and disease. In addition, iPiDA-sHN and DFL-PiDA outperform iPiDi-PUL, which show that neural networks are effective than traditional models.

## Case study

In this part, the prediction performance of MC-GVAE was further evaluated, three diseases, renal cell carcinoma, Alzheimer's disease, and cardiovascular diseases with the largest number of associations in the association matrix were selected as case studies. Specifically, the association information of three diseases-related was firstly removed from the benchmark dataset, and the data remained was employed as the positive samples for training. Then, three diseases-related associations were used as positive samples in each prediction process, and the negative samples from unknown associations of three diseases with other piRNAs. Finally, the prediction scores between three diseases and other known piRNAs were obtained, respectively, and the top 20 piRNAs-related with the highest prediction score were selected for analysis.

Tables 6–8 shows the top 20 renal cell carcinoma-related, the top 20 predicted Alzheimer's diseases-related, and the top 20 predicted cardio-vascular diseases-related piRNAs which achieves the high scores of disease-associated prediction, respectively. As

**Table 6 Top 20 predicted renal cell carcinoma-related piRNAs.**

| Rank | piRNA ID | Evidence | Rank | piRNA ID | Evidence |
|---|---|---|---|---|---|
| 1 | piR-hsa-27703 | piRDisease | 11 | piR-hsa-7770 | piRDisease |
| 2 | piR-hsa-8086 | piRDisease | 12 | piR-hsa-16646 | piRDisease |
| 3 | piR-hsa-3577 | piRDisease | 13 | piR-hsa-26809 | piRDisease |
| 4 | piR-hsa-20320 | unconfirmed | 14 | piR-hsa-23919 | piRDisease |
| 5 | piR-hsa-29714 | piRDisease | 15 | piR-hsa-13300 | piRDisease |
| 6 | piR-hsa-8104 | piRDisease | 16 | piR-hsa-15195 | piRDisease |
| 7 | piR-hsa-27148 | piRDisease | 17 | piR-hsa-15696 | piRDisease |
| 8 | piR-hsa-4269 | piRDisease | 18 | piR-hsa-11906 | piRDisease |
| 9 | piR-hsa-10523 | piRDisease | 19 | piR-hsa-12955 | piRDisease |
| 10 | piR-hsa-19848 | piRDisease | 20 | piR-hsa-20480 | piRDisease |

**Table 7 Top 20 predicted Alzheimer's diseases-related piRNAs.**

| Rank | piRNA ID | Evidence | Rank | piRNA ID | Evidence |
|---|---|---|---|---|---|
| 1 | piR-hsa-28851 | piRDisease | 11 | piR-hsa-27399 | piRDisease |
| 2 | piR-hsa-24775 | piRDisease | 12 | piR-hsa-19620 | piRDisease |
| 3 | piR-hsa-18393 | Unconfirmed | 13 | piR-hsa-30892 | piRDisease |
| 4 | piR-hsa-15837 | Unconfirmed | 14 | piR-hsa-28188 | piRDisease |
| 5 | piR-hsa-9851 | piRDisease | 15 | piR-hsa-11129 | Unconfirmed |
| 6 | piR-hsa-23209 | *Roy et al. (2017)* | 16 | piR-hsa-1823 | *Roy et al. (2017)* |
| 7 | piR-hsa-24366 | Unconfirmed | 17 | piR-hsa-2107 | *Roy et al. (2017)* |
| 8 | piR-hsa-6147 | piRDisease | 18 | piR-hsa-13013 | Unconfirmed |
| 9 | piR-hsa-19012 | Unconfirmed | 19 | piR-hsa-15023 | *Roy et al. (2017)* |
| 10 | piR-hsa-1849 | *Roy et al. (2017)* | 20 | piR-hsa-29716 | Unconfirmed |

**Table 8 Top 20 predicted cardio-vascular diseases-related piRNAs.**

| Rank | piRNA ID | Evidence | Rank | piRNA ID | Evidence |
|---|---|---|---|---|---|
| 1 | piR-hsa-21532 | piRDisease | 11 | piR-hsa-29617 | piRDisease |
| 2 | piR-hsa-27477 | piRDisease | 12 | piR-hsa-18847 | piRDisease |
| 3 | piR-hsa-170 | piRDisease | 13 | piR-hsa-12656 | piRDisease |
| 4 | piR-hsa-29225 | piRDisease | 14 | piR-hsa-17388 | piRDisease |
| 5 | piR-hsa-14359 | piRDisease | 15 | piR-hsa-23376 | piRDisease |
| 6 | piR-hsa-21348 | piRDisease | 16 | piR-hsa-8322 | piRDisease |
| 7 | piR-hsa-28877 | piRDisease | 17 | piR-hsa-12634 | piRDisease |
| 8 | piR-hsa-12176 | piRDisease | 18 | piR-hsa-22176 | piRDisease |
| 9 | piR-hsa-3790 | piRDisease | 19 | piR-hsa-1053 | piRDisease |
| 10 | piR-hsa-13468 | piRDisease | 20 | piR-hsa-18939 | piRDisease |

shown in Table 6, only one association is still unconfirmed in renal cell carcinoma. In Table 7, eight piRNA candidates of the top 20 in Alzheimer's disease are verified by piRDisease v1.0 database, and the other five candidates' piRNAs are confirmed by published literature. However, seven piRNA candidates with the possible association are unconfirmed. In Table 8, all piRNA candidates in top 20 are predicted for cardio-vascular diseases. In other words, those piRNA-disease associations predicted only according to the known knowledge by our model are verified by biological experiments. It indicates that iPiDA-VAGE is effective in piRNA-disease association prediction.

## CONCLUSIONS

Being a significant biomarker for the diagnosis of diseases, piRNA has attracted much attention of academic community. Compared to traditional biological experiments, computational methods are more effective to find potential associations between piRNAs and diseases. In this article, a novel computational model named MC-GVAE was proposed to predict disease-related piRNAs based on a multi-channel graph VAE. An end-to-end computational framework was adopted, and four similarity networks were employed in order to learn latent association knowledge. For homogeneous networks, a multi-channel method was proposed to integrate them for learning. The experimental results showed that the proposed model was effective to predict piRNA-disease association.

Overall, the advantage of MC-GVAE lies in two folds: (1) it use a graph variational autoencoder, which helps to learn feature representations over graphs; (2) it use a multi-channel method, which helps to capture semantic relations of multiple aspects. On the other hand, there are also potential limitations of MC-GVAE: (1) it depends on rich and high-quality data, which are not always available; (2) it is based on the assumption of static piRNA-disease associations like other GNN-based models, whereas the knowledge could be updated for new biological findings; (3) it lacks the ability of casual inference, which is useful for some applications.

There are some directions for the further improvement of this model. Firstly, the similarity knowledge derived from computational methods, such as sequence similarity, disease semantic similarity and GIP kernel similarity does not always accurate, and the incorrect knowledge maybe noisy to the model. Therefore, more accurate knowledge from biological experiment evidence is needed, such as position-specific information of piRNA sequence thermodynamic and physicochemical properties of piRNAs *etc*. Secondly, there are few ground-truth negative samples in the current knowledge bases of piRNA, and pseudo negative samples *via* random selection may lead to poor robustness of the model. In the furure work, more reasonable sampling methods, such as positive unlabeled learning (PUL), or non-sampling learning methods, could be adopted to make a more robust model. Thirdly, considering that our model is independent from concrete RNA sequences and diseases, it can be easily transferred to other prediction tasks of RNA-disease association. To this end, more experiments with different ncRNA databases will be made to validated the performance of the proposed model in the future work.

### Funding

This work is supported by Major Project of Philosophy and Social Sciences of the Ministry of Education (No. 21JDA050), the Research Fund of National Language Commission (No.YB145-2), the Guangdong Education Department Project Foundation (No. 2017KTSCX064), Guangdong Philosophy and Social Sciences Foundation (No. GD20XZY01), the Guangdong University of Foreign Studies Project Foundation (Nos. LAI202305, LEC2019ZBKT002, LEC2022ZBKT005), the Guangzhou Science and Technology Project Foundation (202201010717), the Hainan Natural Science Foundation (Nos. 620QN282, 621MS054) and the China Ministry of Education Foundation (No. 21YJC740058). The funders had no role in study design, data collection and analysis, decision to publish, or preparation of the manuscript.

### Grant Disclosures

The following grant information was disclosed by the authors:
Major Project of Philosophy and Social Sciences of the Ministry of Education: 21JDA050.
Research Fund of National Language Commission: YB145-2.
Guangdong Education Department Project Foundation: 2017KTSCX064.
Guangdong Philosophy and Social Sciences Foundation: GD20XZY01.
Guangdong University of Foreign Studies Project Foundation: LAI202305, LEC2019ZBKT002 and LEC2022ZBKT005.
Guangzhou Science and Technology Project Foundation: 202201010717.
Hainan Natural Science Foundation: 620QN282, 621MS054.
China Ministry of Education Foundation: 21YJC740058.

### Competing Interests

The authors declare that they have no competing interests.

### Author Contributions

- Wei Sun conceived and designed the experiments, performed the experiments, analyzed the data, performed the computation work, prepared figures and/or tables, authored or reviewed drafts of the article, and approved the final draft.
- Chang Guo conceived and designed the experiments, performed the experiments, analyzed the data, performed the computation work, prepared figures and/or tables, authored or reviewed drafts of the article, and approved the final draft.
- Jing Wan analyzed the data, authored or reviewed drafts of the article, and approved the final draft.
- Han Ren conceived and designed the experiments, analyzed the data, authored or reviewed drafts of the article, and approved the final draft.

### Data Availability

The code and the dataset are available in the Supplemental Files.

## Supplemental Information

Supplemental information for this article can be found online at http://dx.doi.org/10.7717/peerj-cs.2216#supplemental-information.

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
