# Peer review of "piRNA-disease association prediction based on multi-channel graph variational autoencoder"

_PeerJ Computer Science, doi:10.7717/peerj-cs.2216_

## Round 0.1 · original submission · Major Revisions

The reviewers have substantial concerns about this manuscript. The authors should provide point-to-point responses to address all the concerns and provide a revised manuscript with the revised parts being marked in different color.

Reviewer 1 ·

Basic reporting

1. The manuscript contains several instances of unclear sentences. For example, Line 268-270. "In order to verify that the GCN encoder module can effectively obtain piRNA and disease feature representation, Firstly, w a neural network encoder-based Variational Autoencoder (VAE) without GCN was designed." This sentence is not only grammatically incorrect but also lacks clarity. It should be revised for grammatical accuracy and to clearly convey the intended message.

2. The authors have provided a comprehensive background and cited relevant literature, which helps put their work within the broader field. However, it would be beneficial to include a more detailed discussion on how their method compares to existing models, especially in terms of the advantages and potential limitations of MC-GVAE.

Experimental design

1.More details are needed on the piRDisease database version used and how the benchmark dataset was constructed. Were any data preprocessing steps applied?

2. The design does not leverage available piRNA knowledge bases for validation. Evaluation is limited to cross validation on piRDisease database, rather than independent biological validation.

3. While the general methodology is described, more details are needed for reproducibility. Hyperparameters, model architecture specifics, training procedures, and piRNA sequence feature extraction require more elaboration.

Validity of the findings

1. For the validation of their model, the authors could consider comparing their predictions not only with piRDisease v1.0 but also with independent datasets to demonstrate the generalizability of their model.

2. The ablation study is a good approach, but the manuscript could benefit from a more detailed discussion of how each component contributes to the overall performance, possibly including statistical significance testing.

3. The manuscript compares MC-GVAE with other methods like iPiDi-PUL, iPiDA-sHN, and DFL-PiDA. However, the basis for choosing these specific methods for comparison is not explained. Clarifying why these particular methods were chosen for comparison would be helpful.

Reviewer 2 ·

Basic reporting

The research paper titled "piRNA-disease association prediction based on multi-channel graph variational autoencoder" presents a novel computational method for predicting piRNA-disease associations using a multi-channel graph variational autoencoder (MC-GVAE). The method integrates four types of similarity networks for piRNAs and diseases, derived from piRNA sequences, disease semantics, piRNA Gaussian Interaction Profile (GIP) kernel, and disease GIP kernel. These networks are modeled using a graph VAE framework to learn low-dimensional and informative feature representations. A multi-channel method is then used to fuse the feature representations, and a three-layer neural network classifier is applied to predict potential associations. The method achieves state-of-the-art performance on a benchmark dataset, demonstrating its effectiveness and superiority in piRNA-disease association prediction. The topic of the paper is significant, however, I have a few concerns that would like the authors to address:

1.Could this association prediction be accomplished using a Transformer model?

2.In association prediction task, what is the advantage of the graph VAE compared with Transformer model?
Please specify details in DISCUSSION.

3.How about the explainability of the graph VAE in this task compared with Transformer?

Ying, C., Cai, T., Luo, S., Zheng, S., Ke, G., He, D., ... & Liu, T. Y. (2021). Do transformers really perform badly for graph representation?. Advances in Neural Information Processing Systems, 34, 28877-28888.

Experimental design

As above

Validity of the findings

As above

Additional comments

As above

Reviewer 3 ·

Basic reporting

This paper develops an algorithm of prediction using a multi-channel graph variational autoencoder and applying the algorithm on piRNA-disease association. Overall the idea was well motivated and the methodological design is reasonable. However, here are a few things can be improved.
1. Typo: double space in line 11; hyphen in line 27; hyphen in line 44; hyphen in line 52; dash in line 66.
2. Figures and Tables: Figure index is not correct for Figure 2-5; table title is not correct and not well-explained for Table 4-6. The meaning of symbol (blue dots and green diamonds) is confusing in Figure 1.
3. Insufficient explanation in rationale: in line 71, Is any evidence available to prove that treating similarity features as independent will lead a worse result?
4. Missing reference: in line 155, "based on the topology of the known association knowledge
of piRNAs and diseases and the interaction profile corresponding to each piRNA and disease". Please fully explain this and cite the corresponding reference.
5. Results: in line 293, the authors stated that "the graph VAE is significantly superior to the other state-
294 of-the-art models". However, in Table 2, GVAE didn't have a better Acc than GAT.

Experimental design

1. Unclear about some parts of methods: how is the adjacency matrix constructed and how is it look like? In line 117, how do you "concatenate" the four different feature representations? In line 206, "the aim of the multi-channel method is to model information from multiple independent aspects". However, in Introduction, the authors mentioned a conflict idea that the similarity features are not supposed to be independent.
2. Missing definition: In line 142, what is the definition of "children" here? In line 147, what is the definition of "T(d)"?

Validity of the findings

no comment

Additional comments

Potential improvement:
1. what's the computing time for all methods/variants compared in the manuscript?
2. how are the performance of other methods on the case study?

Reviewer 4 ·

Basic reporting

The manuscript "piRNA-disease association prediction based on multi-channel graph variational autoencoder," by Dr. Sun et al. proposes a novel computational approach for predicting piRNA-disease associations. The authors introduce a multi-channel graph variational autoencoder (MC-GVAE) that integrates various similarity networks for piRNAs and diseases, derived from diverse sources like piRNA sequences, disease semantics, and Gaussian Interaction Profile (GIP) kernels. The manuscript is well-structured, adhering to PeerJ standards, and presents its research in clear, professional English. The authors have supplied relevant raw data and high-quality figures that enhance the understanding of their methodology and findings

Experimental design

The study falls within the scope of the journal and addresses a significant gap in understanding piRNA-disease associations. It presents original research with a well-defined, relevant, and meaningful research question. The methods are thoroughly described, demonstrating rigorous investigation and adherence to high technical and ethical standards. The research utilizes a benchmark dataset from the piRDisease v1.0 database, allowing for reproducibility and validation of the results

Validity of the findings

The manuscript showcases robust and statistically sound data. The MC-GVAE model's effectiveness is demonstrated through comprehensive performance metrics, including an average AUC value of 0.9310 and an AUPR value of 0.9247 under 5-fold cross-validation. The results are linked to the original research question, and the conclusions are well-stated, offering insights into the model's superior performance in piRNA-disease association prediction compared to existing methods

Additional comments

The manuscript is a commendable contribution to the field of computational biology, particularly in the study of piRNA-disease associations. The novel MC-GVAE model addresses existing limitations in the field, such as the challenge of fusing multi-layer similarity information and the need for more complex representation learning methods. However, the authors acknowledge areas for improvement, such as the reliance on computational methods for similarity knowledge and the need for more accurate biological experiment evidence. Future work could explore these aspects to enhance the model's robustness and accuracy.

---

## Round 0.2 · Minor Revisions

There are some remaining minor concerns

Reviewer 2 ·

Basic reporting

I am satisfied the improvement the authors made for this version of paper.

Experimental design

No.

Validity of the findings

No.

Additional comments

For model performance comparison, could you perform the Matthews correlation coefficient (MCC) as the evaluation metrics in Table 2 & 3?

Chicco, D., Jurman, G. The advantages of the Matthews correlation coefficient (MCC) over F1 score and accuracy in binary classification evaluation. BMC Genomics 21, 6 (2020). https://doi.org/10.1186/s12864-019-6413-7

---

## Round 0.3 · accepted · Accept

Reviewers are satisfied with the revisions and I concur to accept this manuscript.

Reviewer 1 ·

Basic reporting

See below

Experimental design

See below

Validity of the findings

See below

Additional comments

After reading the revised manuscript, this article could be accepted for publication now

Reviewer 2 ·

Basic reporting

I am satisfied with the improvement made by the authors. I have no further comments on current version of paper draft.

Experimental design

No.

Validity of the findings

No.

Additional comments

No.